# Pleiotropic Effect of the *compactum* Gene and Its Combined Effects with Other Loci for Spike and Grain-Related Traits in Wheat

**DOI:** 10.3390/plants11141837

**Published:** 2022-07-13

**Authors:** Mingxing Wen, Jiaxuan Su, Chengzhi Jiao, Xu Zhang, Tao Xu, Tong Wang, Xiaoxue Liu, Zongkuan Wang, Li Sun, Chunxia Yuan, Haiyan Wang, Xiue Wang, Jin Xiao

**Affiliations:** 1State Key Laboratory of Crop Genetics and Germplasm Enhancement, Cytogenetics Institute, Nanjing Agricultural University/JCIC-MCP, Nanjing 210095, China; wmxcell2007@163.com (M.W.); 2019101120@njau.edu.cn (J.S.); jiaocz1990@163.com (C.J.); 2018201061@njau.edu.cn (X.Z.); 2017201029@njau.edu.cn (T.X.); twang@njau.edu.cn (T.W.); liuxiaoxue@njau.edu.cn (X.L.); wangzkuan@njau.edu.cn (Z.W.); sunli@njau.edu.cn (L.S.); yuancx@njau.edu.cn (C.Y.); hywang@njau.edu.cn (H.W.); 2Zhenjiang Institute of Agricultural Science, Jurong 212400, China

**Keywords:** *compactum* gene, mapping, pleiotropic, spike, grain, combination

## Abstract

Club wheat (*Triticum aestivum* ssp. *compactum*) with a distinctly compact spike morphology was conditioned by the dominant *compactum* (*C*) locus on chromosome 2D and resulted in a redistribution of spike yield components. The disclosure of the genetic basis of club wheat was a prerequisite for the development of widely adapted, agronomically competitive club wheat cultivars. In this study, we used a recombinant inbred line population derived from a cross between club wheat Hiller and modern cultivar Yangmai 158 to construct a genetic linkage map and identify quantitative trait loci associated with 15 morphological traits. The club allele acted in a semi-dominant manner and the *C* gene was mapped to 370.12–406.29 Mb physical region on the long arm of 2D. Apart from compact spikes, *C* exhibited a pleiotropic effect on ten other agronomic traits, including plant height, three spike-related traits and six grain-related traits. The compact spike phenotype was correlated with decreased grain size and weight, but with an increase in floret fertility and grain number. These pleiotropic effects make club wheat have compatible spike weight with a normal spike from common wheat. The genetic effects of various gene combinations of *C* with four yield-related genes, including *Ppd-D1*, *Vrn-D3*, *Rht-B1b* and *Rht8*, were evaluated. *C* had no epistatic interaction with any of these genes, indicating that their combinations would have an additive effect on other agronomically important traits. Our research provided a theoretical foundation for the potentially effective deployment of *C* gene into modern breeding varieties in combination with other favorable alleles.

## 1. Introduction

Bread wheat (*Triticum aestivum* L.) is one of the most important grain crops in the world, accounting for 20% of calories for human consumption and feeding more than 35% of the world’s population [1]. The improvement in yield and quality is of great importance for meeting the demand of the world’s growing population. Of the agronomic traits, spike density (SD), measured by the ratio of total spikelet number per spike-to-spike length, is an important spike morphological character and crucial determinant for wheat yield. Breeding wheat varieties with long but relatively compact spikes could improve yield via increasing kernel numbers per spike [2]. Three well-known major genes affecting SD are *C*, *Q* and *S1*, which were located on chromosomes 2D, 5A and 3D, respectively [3,4,5]. Both *C* gene from *T. aestivum* ssp. *compactum* and *S1* gene from *T. aestivum* ssp. *sphaerococcum* are responsible for high SD or compact spike. Increasing the gene expression of *Q* allele or its homoeologous allele on 5D (*5Dq*) also contribute to spike compactness [6,7]. New sources with compact spikes have been created from tetraploid and hexaploid wheat and mapped as either allele of each other or orthologous or novel genes (e.g., *s^16219^*, *C^739^*, *C^17648^*, *C^pm^* and *C^p^*) [8,9]. Several wheat *Reduced Height Genes* (*Rht*) also affect the spike development, resulting in a changed SD, including *Rht1*, *Rht8*, *Rht12*, *Rht13*, etc. [10,11,12,13,14]. QTL mapping studies have been conducted for the genome-wide identification of SD associated loci and nearly all the chromosomes were mentioned as involved in the expression of SD [2,15,16,17,18,19,20,21,22]. Therefore, SD is a complex trait and controlled by multiple genes.

*C* gene results in an extremely compact spike, and this pronounced phenotype has established a unique wheat subspecies—*T. aestivum* ssp. *compactum* (club wheat) [23]. Due to the drought and shattering resistance, stiff straw, earliness and competitive yield, club wheat cultivars are still grown commercially but their distribution is limited to the U.S. Pacific Northwest, as well as a few areas of Middle East (Afghanistan, Iran, Pakistan), Asia (Armenia, Turkey) and Europe (Austria, Switzerland) [4,24]. Club wheat has comparable yield to closely related common wheat, but their yield components are different in terms of seed size and number. It has smaller grains but the decrease in size can be compensated by an increase in grain number [25]. Previous studies have mapped *C* gene on chromosome 2D, but its chromosomal arm location remained unascertainable [26]. It was speculated that *C* was located proximally to centromere, which made it an impediment to its isolation [4]. Mutation at *C* locus possibly gave rise to this dominant allele and resulted in the origin of club wheat; thus, the investigation of this gene is important for the taxonomy of wheat species [27].

*C* gene affects important agronomic traits, such as spike dimension, grain number and shape and perhaps other aspects of plant development that are valuable targets for wheat improvement. It is worth the effort to evaluate its performance, especially in the background of modern wheat cultivars. However, there have been relatively few studies involving this gene concerning its precise chromosomal localization, the pleiotropic effects on other agronomic traits and its combined effects with other known loci for spike and grain-related traits. In the present study, we conducted QTL analysis for the grain- and spikelet shape-related traits using a recombination inbred line (RIL) population derived from the cross between club wheat Hiller (HL) and a widely-grown wheat variety Yangmai 158 (YM), with the objectives to: (1) determine the chromosomal arm location of *C*; (2) evaluate its pleiotropic effect on agronomic performance; (3) dissect its combined effects with other loci for spike and grain-related traits for the effective deployment of *C* gene into modern breeding varieties. The study would lay the foundation for further fine mapping and isolation of *C* gene. These obtained results would be informative as a prerequisite for the development of a widely adapted, agronomically competitive club wheat cultivar.

## 2. Results

### 2.1. Phenotypic Analyses

The descriptive statistics comprise mean and range for plant height (PH, cm), tiller number (TN), six spike-related traits and seven spike-related traits measured from the RIL population and their parental lines YM and HL from four individual environmental trials and BLUP values across environments (Appendix A). YM exhibited higher SL, SPN, grain size and TKW than those of HL, while HL showed higher values for PH and SD (Figure 1A,B). Broad-sense heritability was high for SL (98.53%) and SD (98.60%) and low for TN (40.04%). The average coefficients of variation for the above traits were 2.72–38.83%, indicating that they had a high genetic variance. For all traits, significant genotypic effects and genotype by environment effects were detected. RIL population displayed continuous distributions for PH, TN, five spike-related traits (SPN, FSPN, SSPN, KNS, SW), five grain size-related traits (KL, KW, KPS, KAS, KLWR) and TKW, indicating polygenic inheritance control of these traits. By comparison, SL and SD showed a bimodal frequency distribution, indicating that they were controlled by a few major genes and several minor genes (Figure 2 and Appendix A).

F_1_ plants between YM and HL showed relatively compact spikelets, with the SD falling in the middle of the range of parents (Figure 2). In the RIL population, strong transgressing segregations were detected for SD, with trait values of RILs being higher or lower than those of the two parents (Figure 2). RIL lines could be unambiguously classified into two groups, “normal spike” and “compact spike”, according to their spike morphology (Figure 1C) and the SD frequency distribution (Figure 2A–E). The “compact spike” group was distinct from the “normal spike” group, despite the presence of SD variation within the same group. Using the BLUP values, the SD border could be set at 2.80–2.99 spikelets/cm. By this, RILs having SD that varied from 1.49 to 2.80 spikelets/cm were classified as the “normal spike” group and those having SD that varied from 2.99 to 5.26 spikelets/cm were classified as “compact spike” group (Figure 2F). In the RIL population consisting of a total of 240 lines, the segregation of the two groups were 128 “normal spike” lines to 112 “compact spike” lines, fitting a 1:1 ratio (χ^2^ = 1.07, *p* > 0.05). This implied that the compact spike phenotype of HL in the RIL population was controlled by a single semi-dominant allele, presumably the *C* gene.

We further examined the effect of the compact spike (controlled by *C*) on the other 14 yield-related traits. The means of the above traits were compared between the two RIL groups classified as normal and compact spikes, using BLUP data (Table 1) and the data from four-year environments trials (2018–2021) (Appendix A). The obtained results were similar for all the data sets used for analysis. In addition to SD, 12 traits including PH, four spike-related and seven grain-related traits showed significant differences between the two groups. Particularly, significant differences were observed for SL, KL, KPL, KAS and TKW, implying that the compact spike phenotype significantly correlated with grain shape. We observed that plants with compact spikes had smaller but more grains, which led to a compatible SW (2.19 g) in relation to those plants with normal spikes (2.38 g).

To validate the relationships between SD and other traits, correlation coefficients among the examined traits were calculated in the RIL population (Appendix A). Consistent with the above results, SD, predominantly controlled by *C* gene showed significantly positive correlations to KNS, whereas it exhibited negative correlations to PH, SL, SSPN, six grain size and grain weight traits. Compact spikes were related to a decrease in grain weight and grain size, but with an increase in floret fertility and grain number.

### 2.2. Linkage Groups Construction and QTL Detection for SD and Other Morphological Traits at C Locus

A total of 23,497 SNPs was polymorphic between YM and HL. The resultant linkage map consisted of 13,903 SNP markers (1752 bin markers) assigned to 24 linkage groups (Appendix A). The total length of the linkage map was 2037 cM and the average interval distance between adjacent markers was 1.2 cM. Chromosome size ranged from 42.0 cM (chromosome 1D) to 144.3 cM (chromosome 5B) and the number of loci per chromosome ranged from 66 (chromosome 1D) to 1782 (chromosome 2A).

Previous studies have mapped *C* gene to probably the centromere of chromosome 2D [4]. In this study, QTL analysis for SD was conducted to confirm whether the QTL position for compact spike phenotype could correspond to *C* locus. Two QTL for SD were detected on 2D. One QTL was repeatedly detected in all environments with a LOD score of 88.51–111.43 and contributing to 83.67–88.96% of the SD variation in the RIL population (Figure 3, Table 2). The high LOD scores and the large proportion of its contribution to the QTL for SD indicated that this QTL should be the *C* locus. The linkage SNP markers were closely associated with RILs’ SD morphology, which could be grouped as “Normal spike” and “Compact spike” (Appendix A). The above finding supports our prediction that *QSD.nau-2D.1* should be contributed by the *C* gene.

QTL for the other 14 agronomic traits were also detected to identify whether there were QTL present at *C* locus on 2D, using the same genetic map and BLUP values from four environments. In addition to the QTL for SD, QTL for other ten morphological traits, with the exception of TN, KNS SSPN and FSPN, showed significant LOD scores of >3.0 (*p* < 0.05) at *C* locus. The QTLs mapped to the *C* locus had high LOD scores of 8.09–109.09, and their contributions varied from 5.35 to 80.41% (Figure 3, Table 3). The QTL for SL, TKW, KL, KPL and KAS had LOD scores higher than 37.70 and explained 33.23–80.08% of the variation in the RIL population. The HL allele at the *C* locus positively affected SD, whereas it negatively influenced the other 10 agronomic traits.

### 2.3. Mapping QTL for Five Agronomic Traits Located at Genome Regions Other Than the C Locus

The putative QTL for plant height and spike-related traits detected based on the RIL population evaluated in four environments are summarized in Appendix A. Up to 28 additive QTL were detected in multi-environments, including 13 repeatable QTL (twice or more) for the five investigated traits. Of these, including *C* locus, six were major and stable QTL that accounted for more than 10.0% of the phenotypic variance.

Eleven QTL for PH were detected on 2D (4 QTL), 3A (1), 4A (1), 4B (1), 5A (1), 5D (1) and 6A (2). The phenotypic variation explained by individual QTL ranged from 3.00 to 23.69%. *QPH.nau-2D.2* and *QPH.nau-4B.1* are stable and major QTL. They were detected in all three environments and explained 12.36–23.69% of the phenotypic variance. The two alleles for the decreased PH were all from the parent YM. Out of two QTL detected in two environments, *QPH.nau-2D.1* is *C* locus, explaining 5.77–6.18% of the phenotypic variance with the allele for the decreased PH from the parent HL. *QPH.nau-5D* explained 3.03–4.88% of the phenotypic variance with the allele for the decreased PH from the parent YM. The remaining seven QTL were only detected in a single environment. 

Nine QTL for TSPN were distributed on chromosomes 2A (2), 2D (3), 4A (2), 5A (1) and 7D (1). The phenotypic variation explained by individual QTL ranged from 3.36 to 17.26%. *QTSPN.nau-2D.3* and *QTSPN.nau-7D.1* were the primary stable QTL identified in three environments. In particular, *QTSPN.nau-2D.3* contributed to more than 12.52% of the phenotypic variance and its additive effect was negative, indicating that YM contributed to the allele for the decreased TSPN. *QTSPN.nau-7D.1* explained 3.40–8.46% of the phenotypic variance with the allele for the higher TSPN from YM. *QTSPN.nau-2A.1* and *QTSPN.nau-4A.1* were detected in two environments and explained 4.24–10.91% and 3.76–5.73% of the phenotypic variance, respectively. Both the alleles for the higher TSPN were from HL. The remaining five QTL were detected in a single environment.

Four QTL for SL were detected on chromosomes 2D (3) and 6A (1); they explained 1.09–80.54% of the phenotypic variance. *QSL.nau-2D.1* is *C* locus, which was identified in three environments and explained up to 80.54% of the phenotypic variance. The allele for the decreased SL of *QSL.nau-2D.1* was contributed by HL. *QSL.nau-2D.2* could also be stably detected in three environments with a minor genetic effect, and the allele for the decreased SL was from YM. *QSL.nau-2D.3* and *QSL.nau-6A.1* were identified in two environments, explaining 1.25–1.69% and 1.09–1.25% of the phenotypic variance, respectively, and both alleles for the decreased SL were donated by YM.

All the QTL for SD were repeatedly detected in three environments. As mentioned above, SD showed a significantly negative correlation with SL. Out of four QTLs identified for SD, three were in common with that detected for SL. The major *QSD.nau-2D.1*, which explained up to 81.61% of the phenotypic variance, is *C* locus, and the allele for higher SD was contributed by HL. The other two QTLs *QSD.nau-2D.2* and *QSD.nau-6A.1* explained a minor phenotypic variance of 3.02–4.42% and 1.41–1.52%, respectively, and the alleles for the increased SD were from YM. The remaining QTL *QSD.nau-1B* has a limited genetic effect (~1%), with the alleles for the increased SD from YM.

For TKW, a total of 12 QTL on chromosomes 2B (1), 2D (4), 4B (1), 5A (1), 5B (1), 6A (1), 6B (2) and 7D (1) were identified, and the phenotypic variance explained by individual QTL ranged from 2.14 to 33.39%. The major and stable QTL of *QTKW.nau-2D.1*, which was detected in three environments, is *C* locus and explained up to 33.39% of the phenotypic variance. According to the positive additive effects of *QTKW.nau-2D.1*, HL contributed to this allele for the decreased TKW. *QTKW.nau-2D.6* and *QTKW.nau-5A.3* were detected in two environments and explained 3.89–8.50% and 2.42–2.79% of the phenotypic variance, respectively. The remaining nine QTL were detected in a single environment.

### 2.4. Validation of the Relationship of Mapped QTL with Known QTL or Genes

By comparing our mapping results with the previously mapped QTLs or cloned genes summarized (http://wheatomics.sdau.edu.cn/genes/, accessed on 1 March 2022), the known QTLs or causal genes for the loci associated with each trait were predicted (Appendix A) and validated by integrating known genes into the genetic map using corresponding functional markers (Appendix A). 

The *QPH.nau-2D.2*/*QSL.nau-2D.2*/*QSD.nau-2D.2*/*QTKW.nau-2D.2* were located to the same locus, which had pleiotropic effects on PH, SL, SD and TKW, respectively. The allele from YM decreased PH by 14.16%, reduced SL by 3.69% and increased SD by 4.16%, respectively. The allele also reduced TKW by 3.49%, but this was only detected in one environment. The position and effect of this locus is consistent with the previously reported dwarf gene *Rht8* [12]. Marker analysis using *Rht8* linkage marker *gwm261* showed that YM has *Rht8* allele and HL has *rht8* allele (Figure 4A). By integrating *Rht8* into the genetic map, the genetic interval of the QTL (23.42–24.94 Mb) at the distal short arm of 2D overlaps with *Rht8* gene (Appendix A). This validated the possibility that *Rht8* is the causative gene for the effects of this locus. 

*QPH.nau-2D.3*/*QTSPN.nau-2D.3*/*QSL.nau-2D.3* were mapped to the same region in 32.97–35.02 Mb on 2DS, within which *Ppd-D1* gene was located (33.95 Mb). Marker analysis showed that YM has photoperiod insensitive *Ppd-D1a* allele, while HL has photoperiod sensitive *Ppd-D1b* allele (Figure 4A). *Ppd1* has a pleiotropic effect on TSPN, PH and SL. The allele from YM has a relatively high but negative effect on TSPN (averaged 14.76% phenotypic variance). It also slightly reduced PH and SL by 5.28% and 1.47%, respectively. The effects of *QPH.nau-2D.3*/*QTSPN.nau-2D.3*/*QSL.nau-2D.3* are consistent with the previous report for *Ppd-D1*. We integrated *Ppd-D1* gene into the genetic map, and it was located within the QTL interval of *QPH.nau-2D.3*/*QTSPN.nau-2D.3*/*QSL.nau-2D.3* (Appendix A), indicating that *Ppd-D1* contributed to the genetic effect of this locus.

The PH QTL *QPH.nau-4B.1* is mapped within 31.88–40.90 Mb, a genome region close to *Rht-B1* gene (30.86 Mb). The partial sequence of *Rht1* was cloned and sequenced. *Rht-B1* allele in YM had an A-to-T nucleotide substitution, which led to the change of Lys-61 codon (AAG) to a translational stop codon (TAG) (Figure 4B). In addition, *Rht1* gene was located within the *QPH.nau-4B.1* interval (Appendix A). This confirmed our prediction that the largest effect QTL for plant height (*QPH.nau-4B.1*) was the *Rht1*, with *Rht-B1b* allele from YM. 

The QTL *QTSPN.nau-7D.1* was mapped to a 66.54–72.96 Mb interval, which encompassed *VRN-D3* (*FT-D*) on chromosome 7D (68.42 Mb). A 1-bp deletion was identified in the exon 3 of *VRN-D3* in YM (designated as *vrn-D3*) but not in HL (Figure 4C). The deletion in YM led to a frame shift involving 81 amino acids. This polymorphism is reported to be wildly distributed in diverse germplasms and the mutated *VRN-D3* was deduced as a non-functional allele for late flowering. Consistently, our result showed that *vrn-D3* from YM led to delayed flowering time (data not shown). The location of *Vrn-D3* within *QTSPN.nau-7D.1* interval (Appendix A) suggested that *Vrn-D3* is probably the candidate for *QTSPN.nau-7D.1*.

### 2.5. Genetic Effects of Allele Combinations between C and Other Five Known QTLs or Genes

The contribution of the stably detected QTL combinations to each phenotype was assessed by comparing their sums of squares. 

Three loci, *QPH.nau-2D.1* (*C*/*c*), *QPH.nau-2D.2* (*Rht8*/*rht8*) and *QPH.nau-4B.1* (*Rht-B1a*/*Rht-B1b*), were summarized for PH. Six alleles and eight genotypes produced phenotypes between the two parental lines. The genotype with either *Rht8* or *Rht-B1b* could comparably decrease PH by about 10.0% and 16.8%, respectively, and their combination could decrease it by 22.6%, indicating their additive effect on PH (Figure 5A). The genotypes with *C* have the effect of reducing PH only in the presence of *Rht8*, while they do not influence the PH when *Rht8* is absent, indicating that *C* and *Rht8* might work together to control plant height. Combinations of these alleles (*C*_*Rht8*_ *Rht-B1b*), all with effects of reducing PH, were expected to derive the shortest PH (averaged 94.6 cm), and *c*_*rht8*_ *Rht-B1a* to generate the highest (averaged 127.2 cm).

TSPN was mainly affected by *QTSPN.nau-2D.1* (*C*/*c*) and *QTSPN.nau-2D.3* (*Ppd-D1a*/*Ppd-D1b*) and *QTSPN.nau-7D.1* (*Vrn-D3*/*vrn-D3*). *Ppd1* and *Vrn3* genes were involved directly in the regulation of duration of the growing and developmental phase. *C* from HL increased TSPN. *Ppd-D1a* from YM, which confers photoperiod insensitivity and promotes early heading and flowering, decreases TSPN while *vrn-D3* from YM, which delays reproductive growth, increases TSPN. The combinations of these alleles (*C*_ *Ppd-D1b*_*vrn-D3*) derive the most TSPN (averaged 54.5) while *c*_ *Ppd-D1a*_*Vrn-D3* combinations derive the least (averaged 48.2) (Figure 5B). Consistently, genotypes with *C* allele and *vrn-D3* independently generated a higher kernel number per spike (KNS). However, KNS was not influenced by *Ppd-1* alleles (Figure 5C). This indicated a lack of epistatic interactions among these QTLs in controlling SPN and KNS. 

The genetic control of SD and SL was proportionally attributable to *QSD.nau-2D.1*/*QSL.nau-2D.1* (*C*/*c*) and *QSD.nau-2D.2*/*QSL.nau-2D.2* (*Rht8*/*rht8*). *C* from HL and *Rht8* from YM decrease SL and increase SD. RILs possessing *Rht8* significantly decreased SL by 24.28% and increased SD by 16.49%; those possessing the *C* allele significantly decreased SL by 115.59% and increased SD by 104.26%. As expected, those with the combination of *C* and *Rht8* significantly decreased SL by up to 157.46% and increased SD by 138.83% (Figure 5D,E). This indicated a lack of epistatic interactions among the two QTLs on SD and SL, whereby the combination of *C* and *Rht8* derived the highest SD and lowest SL, and vice versa (Figure 5F). Although *nau-2D.1* and *nau-2D.2* were in the same linkage group (chromosome 2D), they showed an independent assortment (*C*_*Rht8*: *c*_*Rht8*: *C*_*rht8*: *c*_*rht8* = 50:67:63:65 which fit a 1:1:1:1 ratio), indicating a long genetic distance between the two loci, with the former expected on the long arm and the latter on the short arm of 2D, respectively. 

## 3. Discussion

*C* gene from club wheat (*T. aestivum* ssp. *compactum*) contributed to a compact spike relative to spikes of common wheat (*T. aestivum* ssp. *aestivum*). As it might impact crop yield due to its effect on spike compactness, rachilla morphology, seed size and number, *C* gene was a potential gene resource in breeding, and there has been an interest in evaluating its possible direct or indirect effects on agronomic performance [24]. The genetic dissection of *C* and the combined effects of *C* with other yield-related genes was also assessed, which would lay a theoretical foundation for the effective deployment of *C* gene into modern breeding varieties in combination with other appropriate genes or favorable alleles.

Club wheat has smaller grains but more seed numbers than common wheat, which makes it comparable to common wheat in its yields [25]. Consistent with our result, the lines with *C* gene have small seed size and very low TKW, probably due to spike compaction, indicating that the small grain trait of club wheat was also controlled by *C* gene. In addition, these lines also have 2.1 more seed numbers per spike than that of lines containing recessive *c* gene, implying that *C* gene also positively regulates seed number. As *C* gene did not affect the TSPN, more grains per spike derived from the *C* allele would be attributed to more grains per spikelet due to their compactness. Moreover, the lines containing *C* have fewer basal sterile spikelets, which could also contribute to more seed numbers. Similarly, the *s1* gene, which confers rigid short culms and dense spikes also bear small but numerous spherical grains [28]. The semi-dwarfing *Rht8* gene, apart from the effect of reducing plant height, contributes to a slightly reduced spike length and week spike compaction, without penalizing TSPN [29]. We have detected in one environment that *Rht8* negatively regulated TKW. Apart from *C* and *Rht8*, we have also identified two other stable QTLs with minor effects on 6A (*QSD.nau-6A.1*) and 1B (*QSD.nau-1B*), respectively, which were related to spike compactness with the positive effects from YM. Since the compactness has a positive relationship with the grain number but a negative effect on seed size and grain weight, the combination of compactum-related genes and other genetic loci for large seed size and high grain weight would lead to spike architectures with improved productivity. 

The pleiotropic effects of *C* have been described in independent studies [4,26], but *C* responsible for the reduced PH was firstly reported in our study. Overall, the lines with *C* and *c* have no statistically significant difference. When *Rht8* is present, the lines bearing *C* showed a significantly reduced PH compared with the lines with *c* allele. Cloning of *Rht8* indicated that it encodes a protein containing a Zinc finger BED-type motif and an RNase H-like domain or RNHL-D1 that regulates plant height via influencing bioactive GA biosynthesis [30,31]. *C* might interact and work together with RNHL-D1 for the reinforcement of this semi-dwarf trait by *Rht8*. This hypothesis would be addressed by the isolation of *C* gene.

Except for *Rht8*, a lack of epistatic interactions between *C* and known genes/QTLs were identified by the genetic effect evaluation of their allele combinations. The photoperiod insensitive *Ppd-D1a* reduces by 3 days the period of the standard ear emergence in winter bread wheat plants, compared to the *Ppd-D1b* [32]. *Ppd-D1a* responsible for decreased PH, as well as reduced TSPN and SL, was attributed to the shortened duration of plant development. The vernalization-sensitive genes *Vrn-1* and *Vrn-3* also have a durable effect on the developmental process and related agronomic traits such as PH, TSPN and SL [33,34]. The semi-dwarf *Rht1* (*RhtB1*) is GA-insensitive and did not bear obvious changes in spike or grain architecture [10]. Different from the above genes, *C* might be GA-sensitive reminiscent of *Rht8* [35] due to the possible relationship between them. It indicated that *C* might exert an independent mechanism in the determination of plant architectures, indicating an additive effect of *C* with other agronomic-related genes. 

*C* gene from club wheat was mapped to 2D chromosome; however, the precise location of *C* was not unambiguous because the markers that were completely linked to *C* or flanked this locus were localized to chromosome bins on either side of the centromere *C* [4,26]. Therefore, genetic linkage of *C* to the centromere was implicated, which made the map-based cloning of this gene a big challenge due to the reduced levels of recombination within this region. The availability of various high-quality wheat genome reference enables the exact allocation of mapped gene loci. By QTL mapping, *C* gene was mapped to 2D within a 370.12–406.29 Mb interval, which is a little far away from the predicted centromere region in 264.4–272.5 Mb interval [36]. Our results suggested that *C* gene was not as near to the centromere, which increased the possibility of isolating *C* gene through a map-based cloning method.

## 4. Materials and Methods

### 4.1. Plant Materials

Yangmai 158 (YM), which has a normal spike shape and spike density, is a widely growing wheat variety in the middle and lower reaches of the Yangtze River in China. Club wheat Hiller (Reg. no. CV-871, PI 587026), which has a compact spike shape and extremely high spike density, is an ancient hexaploid soft white winter wheat cultivar [37]. Two hundred and forty F_2:11_ RILs from the cross YM/HL were used for QTL mapping. The agronomic traits of RILs and two parents were evaluated in four environments at Zhenjiang (Jiangsu) in 2017–2018 (E1), 2018–2019 (E2), 2019–2020 (E3) and 2020–2021 (E4). The experimental design randomized complete blocks with two replications. Each plot comprised two 1 m rows spaced 25 cm apart, with 30 seeds in each row. All the field trials were managed according to local practices.

### 4.2. Phenotypic Evaluation and Statistical Analysis

At maturity, 10 main spikes in the middle of each row were selected from each plot to measure the plant height (PH, cm), tiller number (TN), six spike-related traits [including spike length (SL, cm), spikelet density (SD), spike weight (SW, g), total spikelet number per spike (SPN), fertile spikelet number per spike (FSPN), basal sterile spikelet number per spike (SSPN)] and seven grain-related traits [including number of kernel per spike (KNS), kernel length (KL, mm), kernel width (KW, mm), kernel length-to-width ratio (KLWR), thousand kernel weight (TKW, g), kernel area size (KAS, mm^2^), kernel perimeter length (KPL, mm)]. For each line, the mean values of each trait were calculated across two replicates in each environment. Basic phenotypic statistical analyses and correlation coefficients among traits were conducted using software SPSS version 20.0. The best linear unbiased prediction (BLUP) of target traits in different environments [38] was obtained using SAS V8.0 (SAS Institute, Cary, NC, USA). The broad-sense heritability (*H_b_*^2^) for each trait was estimated using R software following the formula *H_b_*^2^ = *σ*^2^*_g_*/(*σ*^2^*_g_* + *σ*^2^*_ge_*/*n* + *σ*^2^/*nr*), where *σ*^2^*_g_* is the genotypic effect, *σ*^2^*_ge_* is the genotype by environmental effect, *σ*^2^ is the residual error, *n* is the number of environments and *r* is the number of replicates.

### 4.3. Genetic Map Construction and QTL Mapping

DNA was extracted from young leaves of RILs and parents using the CTAB (hexadecyltrimethy ammonium bromide) method. DNA integrity and quantity were checked and confirmed. The DNA of each line and two parents were hybridized on the wheat 55 K SNP array containing 53,063 markers by CapitalBio Technology Co. Ltd. (Beijing, China). The retained markers were analyzed using the BIN function of IciMapping 4.1 (https://isbreedingen.caas.cn/news1/software22/294513.htm, accessed on 1 June 2021) based on their segregation patterns in the RIL population, with the parameters of ‘Missing Rates’ and ‘Distortion Value’ being set as 20 and 0.01, respectively. The groups of bin markers were ordered in JoinMap 4.0 by Kosambi mapping function with the LOD ≥3.0 after preliminary analysis of SNPs with LOD scores ranging from 2 to 10. The genetic maps were drawn with MapChart 2.2 (http://www.biometris.nl/uk/Software/MapChart/, accessed on 1 June 2021). 

QTL mapping in each environment was performed using the IciMapping 4.1 in the Biparental Populations (BIP) module with the inclusive composite interval mapping (ICIM) and LOD score values ≥ 3.0. QTLs were named according to the rules of International Rules of Genetic Nomenclature (http://wheat.pw.Usda.Gov/ggpages/wgc/98/Intro.htm, accessed on 1 June 2021). ‘nau’ represents Nanjing Agricultural University.

### 4.4. Validation of Known Loci or Genes

The alleles of known genes for agronomic traits, i.e., *Rht8* for plant height, *Vrn-D3* for vernalization and *Ppd*-*D1* for photoperiod in genotypes YM and HL, were determined using PCR-based methods. The primer sets were listed in Appendix A. 

SSR marker *Xgwm261* was used as a diagnostic marker for *Rht8*. The 192 bp allele at the *Xgwm261* locus was designated as *Rht8* and a smaller 174 bp allele was *rht8* [35].

Three primers for *Ppd*-*D1*, namely *Ppd-D1*_*F1*, *Ppd-D1*_*R1* and *Ppd-D1*_*R2*, were designed [39] to identify the deletion of 2089 bp in the upstream region of *Ppd-D1a*. The expected PCR product sizes were 288 bp from *Ppd-D1a* and 415 bp from *Ppd-D1b*, photoperiod-sensitive allele without deletion of 2089 bp, respectively.

The alleles of *Rht-B1* and *Vrn-D3* in genotypes YM and HL were determined by homologous gene cloning. For *Rht-B1*, *Rht-B1b* allele contained single nucleotide substitutions that introduce premature stop codons in the N-terminal coding region [40]. Two primers, *Rht-B1-F* and *Rht-B1-R* flanking the SNP region, were designed in this study.

For *Vrn-D3*, a single polymorphism was reported by Bonnin [41], consisting of an insertion–deletion of one G in a poly G located within exon 3. Two primers, namely *Vrn-D3-F* and *Vrn-D3-R* flanking the SNP region, were designed in this study. 

Two functional markers (*M-Rht-B1* and *M-Vrn-D3*) was used for amplification of *Rht-B1* and *Vrn-D3* alleles in the 240 RILs, respectively [42,43].

## Figures and Tables

**Figure 1 plants-11-01837-f001:**
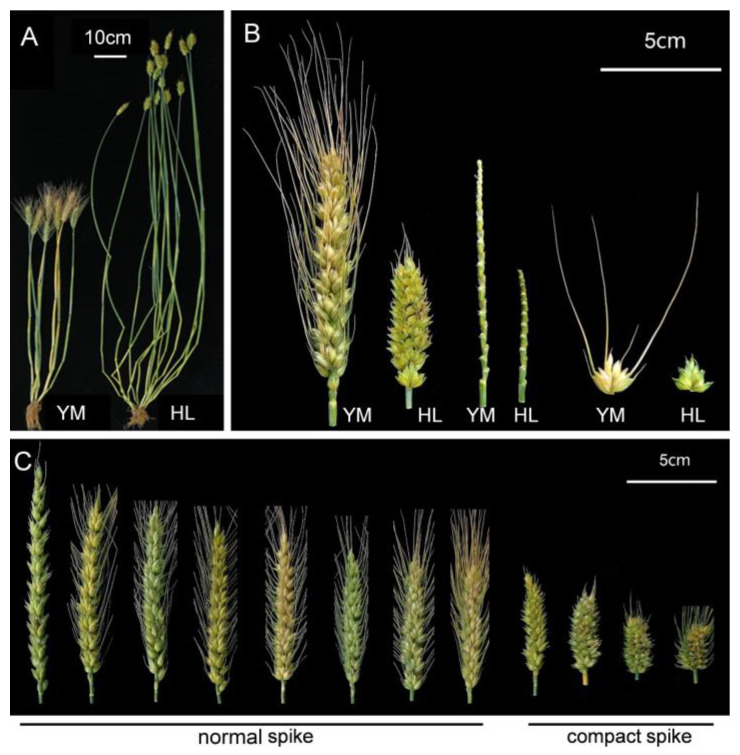
Phenotypes of the parents Yangmai 158 (YM) and *Triticum compactum* Hiller (HL) and their derived representative RILs. (**A**) Plant height of parents; (**B**) Spikes, rachis and spikelets of parents; (**C**) Representative RILs with normal and compact spike.

**Figure 2 plants-11-01837-f002:**
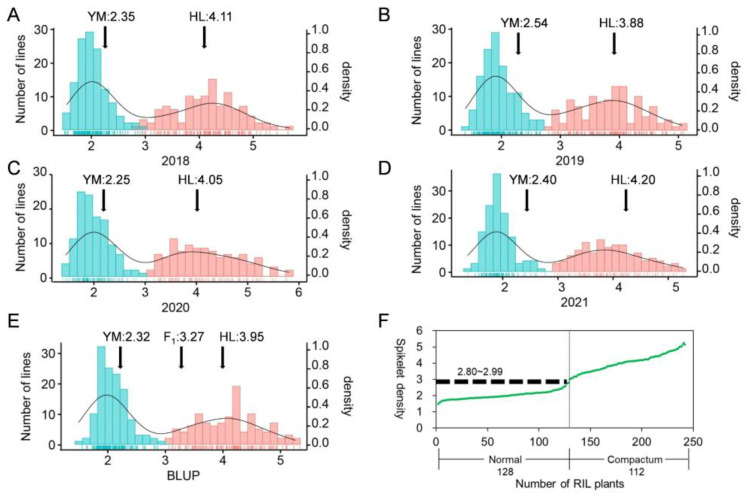
Spike density frequency distribution of Yangmai 158 (YM) × *Triticum compactum* Hiller (HL) RILs using data from each of four-year environments ((**A**–**D**), 2018–2021) and BLUP values across environments (**E**) and classification of RILs according to their SD (**F**). Blue: normal spike group; red: compact spike group. Spike density was measured by the ratio of total spikelet number per spike-to-spike length.

**Figure 3 plants-11-01837-f003:**
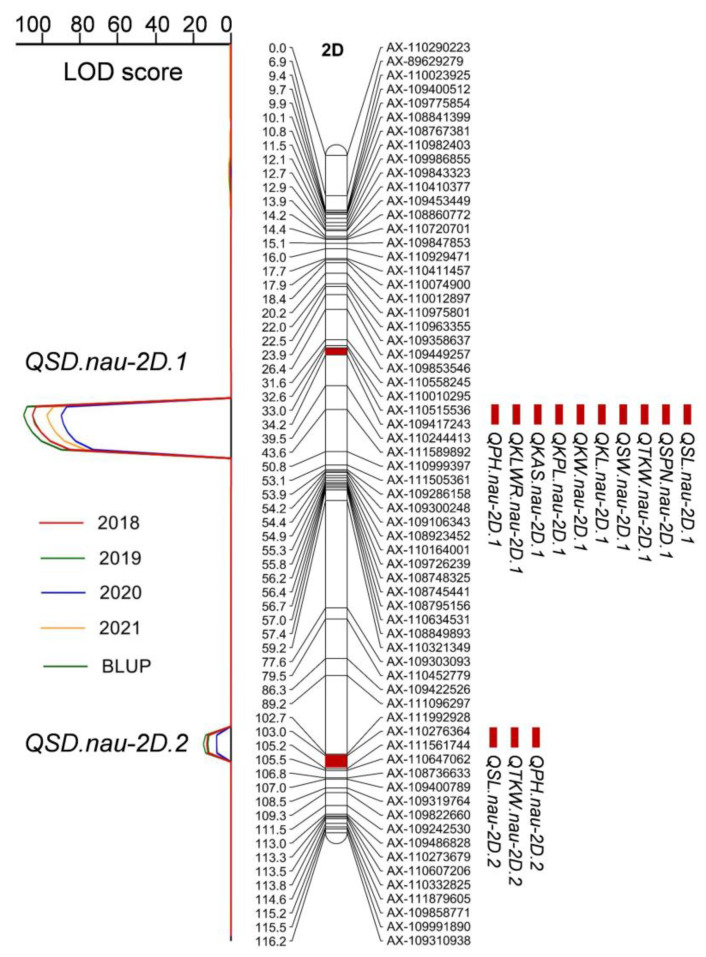
QTL mapping of the morphological traits on chromosome 2D. Middle: a linkage map of chromosome 2D. Genetic distances are represented in centimorgans on the left and SNP markers were on the right of chromosome 2D. Left: QTL likelihood curves of LOD scores (>3.0) based on spikelet density (SD) from four environments during years 2018–2021 and their BLUP values. Right: chromosomal positions of identified QTL for the plant height, grain and spike-related traits on chromosome 2D mapped to the same regions with *QSD.nau-2D.1* and *QSD.nau-2D.2*. BLUP is based on the values from four years’ environmental trials (2018–2021). Abbreviations: PH, plant height; TN, tiller number; SL, spike length; SD, spikelet density; SW, spike weight; SPN, total spikelet number per spike; FSPN, fertile spikelet number per spike; SSPN, basal sterile spikelet number per spike; KNS, number of kernels per spike; KL, kernel length; KW, kernel width; KLWR, kernel length-to-width ratio; TKW, thousand kernel weight; KAS, kernel area size; KPL, kernel perimeter length.

**Figure 4 plants-11-01837-f004:**
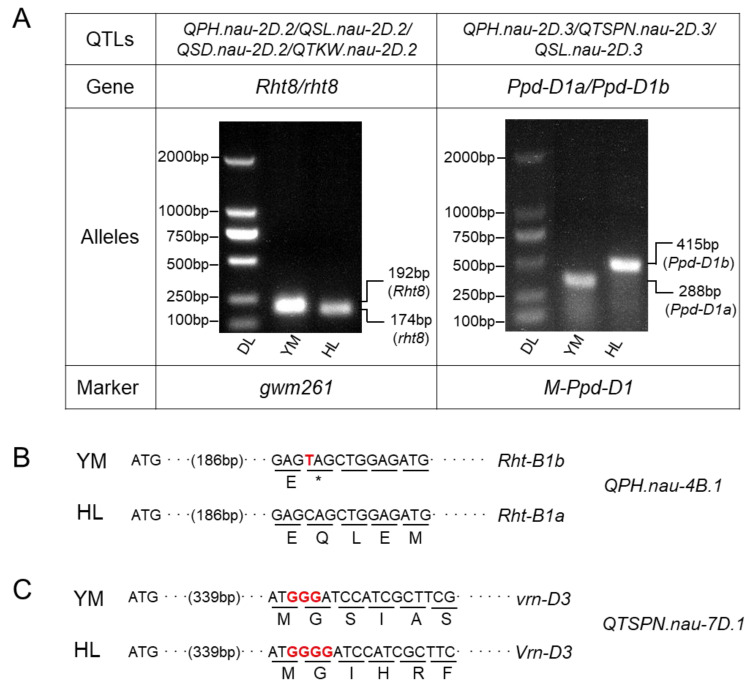
Allele composition of *Rht8*, *PPD-D1* (**A**), *Rht-B1* (**B**) and *Vrn-D3* (**C**) genes in YM and HL. YM: Yangmai158; HL: *Triticum compactum* Hiller; DL: DNA marker.

**Figure 5 plants-11-01837-f005:**
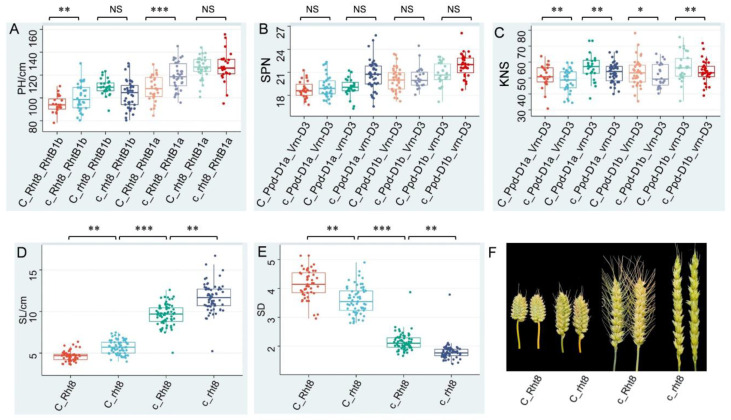
Genetic effects of various gene combinations between *C* and plant height and four spike and grain-related traits. (**A**) The combined effects of *C*, *Rht8* and *Rht1* on plant height (PH); (**B**) The combined effects of *C*, *PPD-D1* and *Vrn-D3* on total spikelet number per spike (SPN); (**C**) The combined effects of *C*, *PPD-D1* and *Vrn-D3* on number of kernels per spike (KNS); (**D**) The combined effects of *C* and *Rht8* on spike length (SL); (**E**) The combined effects of *C* and *Rht8* on spike density (SD); (**F**) The spike morphology of *C* and *Rht8* combinations. Traits that show significant mean difference between genotypes are indicated by * *p* < 0.05, ** *p* < 0.01 or *** *p* < 0.001 (Student’s *t* test). NS is “Non-significance”.

**Table 1 plants-11-01837-t001:** The statistics of 15 investigated traits in RIL population derived from the cross Yangmai 158 (YM) × *Triticum compactum* Hiller (HL).

Traits *	Parents			RILs		
	YM	HL	*p*-Value	Normal Spike	Compact Spike	*p*-Value
PH (cm)	84.99 ± 4.34	130.58 ± 5.72	3.36 × 10^27^ ***	119.01 ± 14.50	114.89 ± 13.80	0.026 *
TN	4.05 ± 1.50	5.75 ± 2.69	0.018 *	6.98 ± 0.19	7.01 ± 0.19	0.249
SL (cm)	8.50 ± 0.59	4.46 ± 0.39	1.32 × 10^25^ ***	10.72 ± 1.49	5.23 ± 0.87	6.53 × 10^94^ ***
SD	2.32 ± 0.14	3.95 ± 0.32	2.30 × 10^22^ ***	2.02 ± 0.25	4.04 ± 0.54	0 ***
SW (g)	2.51 ± 0.40	1.55 ± 0.41	5.19 × 10^09^ ***	2.38 ± 0.26	2.19 ± 0.28	6.37 × 10^08^ ***
SPN	19.65 ± 0.75	17.50 ± 1.05	5.89 × 10^09^ ***	20.89 ± 1.39	20.32 ± 1.23	0.001 **
FSPN	19.35 ± 0.99	17.15 ± 1.35	8.18 × 10^07^ ***	19.84 ± 1.18	19.69 ± 1.06	0.312
SSPN	1.00 ± 0.79	0.50 ± 0.76	0.049 *	1.05 ± 0.51	0.63 ± 0.37	4.66 × 10^12^ ***
KNS	53.70 ± 7.36	45.45 ± 8.85	0.003 **	55.32 ± 4.72	57.32 ± 5.66	0.003 **
KL (mm)	6.50 ± 0.12	5.83 ± 0.07	2.86 × 10^07^ ***	6.55 ± 0.31	6.07 ± 0.29	8.97 × 10^27^ ***
KW (mm)	3.37 ± 0.10	3.00 ± 0.04	5.33 × 10^06^ ***	3.31 ± 0.13	3.22 ± 0.15	2.27 × 10^06^ ***
KLWR	1.95 ± 0.05	2.00 ± 0.02	0.054	2.01 ± 0.07	1.93 ± 0.08	1.91 × 10^14^ ***
KPL (mm)	17.11 ± 0.31	15.29 ± 0.15	1.58 × 10^07^ ***	16.89 ± 0.71	15.79 ± 0.68	4.71 × 10^27^ ***
KAS (mm^2^)	16.29 ± 0.73	12.27 ± 0.33	2.25 × 10^07^ ***	16.10 ± 1.30	14.22 ± 1.23	1.40 × 10^24^ ***
TKW (g)	47.47 ± 9.57	34.89 ± 10.10	2.50 × 10^04^ ***	43.93 ± 4.19	37.75 ± 3.61	9.15 × 10^27^ ***

Notes: The descriptive statistics comprise mean and range for 15 traits measured from the RIL population and their parental lines YM and HL in BLUP values from four environments trials during years 2018–2021. N = 128 with normal spike density (1.49–2.80 spikelets/cm), N = 112 with a compact spike (2.99–5.26 spikelets/cm). Traits that show significant mean difference between genotypes are indicated by * *p* < 0.05, ** *p* < 0.01 or *** *p* < 0.001 (Student’s *t* test). * Abbreviations: PH, plant height; TN, tiller number; SL, spike length; SD, spikelet density; SW, spike weight; SPN, total spikelet number per spike; FSPN, fertile spikelet number per spike; SSPN, basal sterile spikelet number per spike; KNS, number of kernels per spike; KL, kernel length; KW, kernel width; KLWR, kernel length-to-width ratio; TKW, thousand kernel weight; KAS, kernel area size; KPL, kernel perimeter length.

**Table 2 plants-11-01837-t002:** Summary of QTL on chromosome 2D for spike density trait from four environments during years 2018–2021 and their BLUP values in the RIL population.

Trials	QTL	LOD Score	Marker Interval	Physical Distance (Mb)	Additive Effect	Contribution (%)
2018	*QSD.nau-2D.1*	116.53	AX-110515536-AX-109417243	370.1–406.3	−1.09	81.61
2018	*QSD.nau-2D.2*	17.15	AX-110276364-AX-111561744	23.4–24.9	0.25	4.42
2019	*QSD.nau-2D.1*	114.17	AX-110515536-AX-109417243	370.1–406.3	−0.98	80.57
2019	*QSD.nau-2D.2*	18.12	AX-110276364-AX-111561744	23.4–24.9	0.24	5.03
2020	*QSD.nau-2D.1*	93.71	AX-110515536-AX-109417243	370.1–406.3	−1.11	77.82
2020	*QSD.nau-2D.2*	9.16	AX-110276364-AX-111561744	23.4–24.9	0.22	3.02
2021	*QSD.nau-2D.1*	101.44	AX-110515536-AX-109417243	370.1–406.3	−1.04	79.68
2021	*QSD.nau-2D.2*	9.10	AX-110276364-AX-111561744	23.4–24.9	0.20	2.83
BLUP	*QSD.nau-2D.1*	119.61	AX-110515536-AX-109417243	370.1–406.3	−1.04	82.35
BLUP	*QSD.nau-2D.2*	15.60	AX-110276364-AX-111561744	23.4–24.9	0.23	3.96

**Table 3 plants-11-01837-t003:** Summary of QTL for PH and grain and spike-related traits mapped to the same regions with *QSD.nau-2D.1* and *QSD.nau-2D.2* on chromosome 2D in the RIL population using BLUP values from four environments (2018–2021).

Trait *	QTL	LOD Score	Marker Interval	Physical Distance (Mb)	Additive Effect	Contribution (%)
SPN	*QSPN.nau-2D.1*	11.37	AX-109417243-AX-110515536	370.1–406.3	0.57	11.00
SL	*QSL.nau-2D.1*	106.61	AX-109417243-AX-110515536	370.1–406.3	2.89	80.08
TKW	*QTKW.nau-2D.1*	37.91	AX-109417243-AX-110515536	370.1–406.3	3.64	30.46
SW	*QSW.nau-2D.1*	8.58	AX-109417243-AX-110515536	370.1–406.3	0.12	9.51
KL	*QKL.nau-2D.1*	47.78	AX-109417243-AX-110515536	370.1–406.3	0.28	37.76
KW	*QKW.nau-2D.1*	10.61	AX-109417243-AX-110515536	370.1–406.3	0.06	11.80
KPL	*QKPL.nau-2D.1*	42.38	AX-109417243-AX-110515536	370.1–406.3	0.61	38.80
KAS	*QKAS.nau-2D.1*	37.70	AX-109417243-AX-110515536	370.1–406.3	1.10	33.23
KLWR	*QKLWR.nau-2D.1*	24.88	AX-109417243-AX-110515536	370.1–406.3	0.05	19.68
PH	*QPH.nau-2D.1*	8.09	AX-109417243-AX-110515536	370.1–406.3	4.20	5.35
SL	*QSL.nau-2D.2*	11.38	AX-111561744-AX-110276364	23.4–24.9	−0.52	2.27
TKW	*QTKW.nau-2D.2*	32.28	AX-111561744-AX-110276364	23.4–24.9	−2.40	12.90
PH	*QPH.nau-2D.2*	25.30	AX-111561744-AX-110276364	23.4–24.9	−6.11	13.42

* Abbreviations: PH, plant height; SL, spike length; SD, spikelet density; SW, spike weight; SPN, total spikelet number per spike; KL, kernel length; KW, kernel width; KLWR, kernel length-to-width ratio; TKW, thousand kernel weight; KAS, kernel area size; KPL, kernel perimeter length.

## Data Availability

The data presented in this study are available in the article and the Appendix A.

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
