# Peer review of "Pleiotropic Effect of the compactum Gene and Its Combined Effects with Other Loci for Spike and Grain-Related Traits in Wheat"

_plants, 2022, doi:10.3390/plants11141837_

Round 1
Reviewer 1 Report
The article is very good. All parts are well-written and the literature is relevant and current. The tables and figures are also good. There are only some minor mistakes, which the Authors must correct:
In page 6, line156 the reference must be written by a number. The same must be done in pages 11, line 378 and 12, line 397
The most serious mistake is that they use only two replications and this is not enough to face the soil heterogeneity.
Please also include an abbreviation list to facilitate readers.
Author Response
Response to Reviewer 1 Comments
Point 1: The article is very good. All parts are well-written and the literature is relevant and current. The tables and figures are also good. There are only some minor mistakes, which the Authors must correct:
Response 1: Thank you so much for the reviewer's positive comments and support. We have corrected the mistakes accordingly.
Point 2: In page 6, line156 the reference must be written by a number. The same must be done in pages 11, line 378 and 12, line 397
Response 2: Thanks for pointing out the mistakes. We have revised them accordingly.
Point 3: The most serious mistake is that they use only two replications and this is not enough to face the soil heterogeneity.
Response 3: Yes, we have acknowledged this is the main drawback of our experimental design. To compensate this short-come, we have conducted the experiments in four-year trials and obtained similar results across years, indicating a high repeatability and credibility of our field data.
Point 4: Please also include an abbreviation list to facilitate readers.
Response 4: We have added an abbreviation list at the end of the manuscript.
Reviewer 2 Report
Dear authors,
This work with club wheat with a distinctly compact spike morphology was conditioned by the dominant compactum (C) locus on chromosome 2D and resulted in a redistribution of spike yield components. The compact spike phenotype was correlated with decreased grain size and weight, but with an increase in floret fertility and grain number. This work could be potentially interesting in wheat breeding due to potential effective deployment of C gene into modern breeding varieties.
Some technical mistakes
Pg 2, line 58, you missed comma between Austria and Switzerland
Pg 5, line 142 separate 2,037 and cM, the same applied at line 143 and through manuscript
Pg 6, lines 155-156 should you move this sentence in discussion (with reference)
Pg 6, line 171 please take a look guidance for authors, should you separate % from number or not (some where you do that, somewhere not). Please uniform this through manuscript
Pg 7, lines 195-213 you are using different spaces between rows (please make it uniform through text)
In general, in your manuscript your sections are differently separated (somewhere is smaller separation, somewhere larger). This also applies to titles or subtitles through manuscript.
Author Response
Response to Reviewer 2 Comments
This work with club wheat with a distinctly compact spike morphology was conditioned by the dominant compactum (C) locus on chromosome 2D and resulted in a redistribution of spike yield components. The compact spike phenotype was correlated with decreased grain size and weight, but with an increase in floret fertility and grain number. This work could be potentially interesting in wheat breeding due to potential effective deployment of C gene into modern breeding varieties.
Response: Thank you so much for the positive comments and support.
Some technical mistakes
Point 1: Pg 2, line 58, you missed comma between Austria and Switzerland
Response 1: Thanks for pointing out the mistake. We have revised it accordingly.
Point 2: Pg 5, line 142 separate 2,037 and cM, the same applied at line 143 and through manuscript
Response 2: We have revised them accordingly.
Point 3: Pg 6, lines 155-156 should you move this sentence in discussion (with reference)
Response 3: Thank you for the suggestion. However, after careful consideration, we are afraid this sentence is better to be here. This paragraph focusses on the mapping of high spike density (SD) gene which is controlled by the 2D C gene. In present study using the bi-parental population, four QTL for SD trait were detected, and two were on 2D. Therefore, we need clarify the readers that C gene is on 2D (with reference), before choosing the 2D QTL as C candidates.
Point 4: Pg 6, line 171 please take a look guidance for authors, should you separate % from number or not (some where you do that, somewhere not). Please uniform this through manuscript.
Response 4: We have uniform this through manuscript according to the guidance.
Point 5: Pg 7, lines 195-213 you are using different spaces between rows (please make it uniform through text).In general, in your manuscript your sections are differently separated (somewhere is smaller separation, somewhere larger). This also applies to titles or subtitles through manuscript.
Response 5: Thanks for pointing out the mistakes. We have revised them through manuscript.
Reviewer 3 Report
Thank you for giving me extra time for this review. While this manuscript does a good job of mapping the club gene more accurately, it does not describe the club wheat parent accurately so I don't know what the parent actually was. This needs to be corrected. With this correction, I think the article should be published because it does contribute to the literature on club wheat, which is a bit sparse.

Author Response
Response to Reviewer 3 Comments
Thank you for giving me extra time for this review. While this manuscript does a good job of mapping the club gene more accurately, it does not describe the club wheat parent accurately so I don't know what the parent actually was. This needs to be corrected. With this correction, I think the article should be published because it does contribute to the literature on club wheat, which is a bit sparse.
Response: Thank you so much for the positive comments and support.
In 1999, our lab has introduced six club wheat cultivars, Madsen, stephens, Eltan, Bruehl, Hiller and Rod from Washington State University. We have checked them again and found the club wheat parent was actually not Bruehl, but Eltan with a typical compact spike morphology. We have revised them throughout the MS. Thanks for pointing out the mistakes.
Point 1: Line 42: Chen et al., 2020 The Plant Cell, Vol. 32: 923–934, April 2020, www.plantcell.org 2020 ASPB. is more recent reference for the location of S.
Response 1: We have revised it accordingly and cited the work by Cheng et al., 2020.
Point 2: Line 47: These genes were mapped as alleles of each other, but not as alleles of the C gene.
Response 2: We have revised it accordingly in line 47. Thank you.
Point 3: Table 1 (and others). The abbreviations are only defined in the methods, which is at the end of the document. There are several abbreviations that are not intuitive. Please add a footnote to the tables that defines the abbreviations. Tables and figures should stand by themselves and not require additional referral to the text. As an alternative, it is easier to simply spell out the traits in each table rather than resort to abbreviations.
Response 3: We have added an abbreviation list at the end of our manuscript and added a footnote to the tables.
Point 4: Table 1. I find it interesting that the kernel number per spike for the Bruehl parent was lower than that of the Yangmai parent when club wheats generally have more kernels per spike. In fact, Bruehl, while classified in the grain trade as a club wheat, does not possess typical kernel morphology for club wheat and this was a negative factor in the U.S. grain trade.
Response 4: Thank you for the insight comment. Yangmai 158 is one of the most popular Chinese commercial wheat cultivars in the Middle and Lower Valleys of the Yangtze River and Huang-Huai Valleys with the total planting areas of about 17 Mha. The cultivar has desirable agronomic and grain quality traits. “The abundant kernels per spike” is probably one of the favorable traits for its improvement which is not associated with C. However, in RILs, club wheats group generally has more kernels per spike and was probably due to C gene effect.
As pointed above, we have revised “Bruehl” to “Eltan” throughout the MS.
Point 5: Line 130. I don't understand how the coefficient of variation can be related to the improvement potential. In fact, I'm not sure what the term 'improvement potential' means in this context. CV is dependent on the scoring system used for each trait and is often highly correlated with the scale used for the scoring system. For example, plant height generally has a small CV because it is measured at a long length and has high heritability. But the CV itself is not related to its improvement potential while the heritability is. In fact, one could make the argument that the combination of a higher genetic variance (higher CV) plus high heritability would be more desirable.
Response 5: Thank you for the suggestion. We agree that “improvement potential” is not a proper conclusion here. We have revised it to “high genetic variance”.
Point 6: Line 142. Is this average distance calculated based on the total markers and the map length or based on the number of markers that map to a unique locus?
Response 6: The average distance is calculated based on the total map length (2037cM) and the unique loci (1752 bins). The co-separated SNP markers were deemed as one locus or physical bin.
Point 7: I don't think that the allele Rht-B1b is the cause of your QTL because Bruehl also has that allele, as detected mutliple times with the Rht-B1b markers
Response 7: Thanks for pointing out the mistakes. As pointed above, we have revised “Bruehl” to “Eltan” throughout the MS. In our study, Rht-B1a in Eltan was confirmed by homologous gene cloning (Figure 4).
Point 8: Discussion. We have also remapped the C gene and find it on the long arm in the same general location as you have, more proximal to the centromere. So I think that your general conclusions are probably correct. However, I am concerned about the source of the Bruehl that you used because it does not have the correct locus at the Rht-B1 locus. It would help to obtain another source of Bruehl and run it as a check for other genes to verify that the parent that you used is genetically related.
Response 8: Thank you again for the positive comments and support of our results. We are happy to know the compactum gene goes to the same location. We have revised “Bruehl” to “Eltan” throughout the MS.
Point 9: Also, be careful of conclusions made using a single RIL population. As pointed out in the older papers using NILs for the C gene, Gul and Allan, 1972 and Zwer et all., 1995; the effect of the C gene on traits other than spike compaction is, to some extent, dependent on the population.
Response 9: Thank you for your reminding. We have compared part of our results with those reported from Gul and Allan, 1972 and Zwer et all., 1995. We have obtained similar results using different materials.
Point 10: Line 377: Bruehl is not an ancient hexaploid. While it is a hexaploid, Bruehl is a modern semi-dwarf club wheat cultivar, released as a commercial variety in 2001. There are several other modern semi-dwarf club wheat cultivars including the following: Chukar, Cara, ARS Crescent, Castella, Pritchett, and Cameo. (See Garland-Campbell et al., 2013, 2017, 2021a, 2021b, and Campbell 2005).
Response 10: Thank you for providing us this useful information. We have revised “Bruehl” to “Eltan” throughout the MS.
Reviewer 4 Report
The objectives of the manuscript were to precise map the spike compact gene (C) and the major QTL for spike density (SD) of wheat, and report the pleiotropic effects of C on agronomic and grain yield related traits and its combined effects with other important morpho-physiological loci. A segregant population of 240 recombinant inbred lines, developed from crossing the club wheat Bruehl with the modern cultivar Yangmai 158, was evaluated for 15 traits in one location for four years. A high-density SNP-based genetic map was constructed and used for the QTL analysis. Inclusive composite interval mapping (ICIM) was carried out to identify several QTL associated to the corresponding BLUP values of each examined trait. The effects of a major QTL for spike density detected on chromosome arm 2DS and its pleiotropic effects on different grain and spike traits were reported.
Strength of the manuscript.
- QTL mapping of phenotypic data assessed in four replicated field trials.
- A high density SNP-based map used for QTL mapping.
- Development of gene markers for important agronomic traits.
Limitations of the manuscript.
The information is presented in a disorganized and not simple and direct way to be understood by a competent scientist or reader.
Excessive use of abbreviations in the text, figures and tables that do not allow a fluent reading of the manuscript.
Main comments and suggestions.
The information must be presented in a simple and straightforward way to be understood by the reader.
- Paragraph 2.1. on “Phenotypic analyses” should report the ANOVA of each trait for individual environments and across environments to ascertain significant differences between the RILs of the segregant population (Table could reported in Supplemental Materials).
- Table 1 “The 15 morphological traits in RIL population from the cross YM×TC”. As reported in the caption, the table should report data for the whole population without in this paragraph the separation in the two groups "Normal spike" and "Compact spike". Such classification based only on phenotypic data could be considered arbitrary and should be reported subsequently based also on data of the molecular markers closely associated with the major QTL for spike density (SD) on 2D chromosome.
- Figure 2 could be substituted by a graph on the frequency distribution of spike density. The bimodal distribution could suggest the genetic control by a major gene.
- “Linkage groups construction”. The paragraph should report the construction and description of the genetic map including in the map the examined gene Ppd-D1, Vrn-D1, Vrn-D3, Rht-B1b, and Rht8.
- “QTL detection for all the evaluated traits”. Report in the text and discuss all significant QTLs for all traits at LOD> 3.0. The QTL name in the text should always be complete with the trait abbreviation. Paragraphs 2.4 and 2.5 are not understandable for the reader as QTL names are reported without the character name (for example, what QTL does “nau-2D.2” refer to?).
- Table and discussion on the classification of RILs according to phenotypic data on spike density and on molecular markers closely associated with the major QTL for spike density (SD) on 2D chromosome.
- Pleiotropic effects of SD gene/QTL with morpho-physiological and yield related trait.
“Figure 4. Association of the QTLs with known genes located in the mapped QTL regions.” The figure shows the polymorphisms detected between the alleles of three genes and not the associations with QTLs.
- Genes are not present in the genetic map.
- Relationship of mapped QTLs with known QTLs or genes. Report details on the statistical analysis.
- Report the abbreviations used in Figure S1, Figure 3 and Table 3.
- Discussion section: avoid concepts and data already reported in the results.
Author Response
Response to Reviewer 4 Comments
Main comments and suggestions.
Point 1: The information must be presented in a simple and straightforward way to be understood by the reader.
Paragraph 2.1. on “Phenotypic analyses” should report the ANOVA of each trait for individual environments and across environments to ascertain significant differences between the RILs of the segregant population (Table could report in Supplemental Materials).
Response 1: We have performed ANOVA of each trait for individual environment and across environments, and obtained similar results to that from BLUP data. We have added this results in Supplemental file 1 and in lines 129-131.
Point 2: Table 1 “The 15 morphological traits in RIL population from the cross YM×TC”. As reported in the caption, the table should report data for the whole population without in this paragraph the separation in the two groups "Normal spike" and "Compact spike". Such classification based only on phenotypic data could be considered arbitrary and should be reported subsequently based also on data of the molecular markers closely associated with the major QTL for spike density (SD) on 2D chromosome.
Response 2: Thank you for the technical comments. We have reported data for the whole population in paragraph 4 under “2.1 Phenotypic analyses”. We have moved the paragraph 4 in front in lines 89-96.
Because of the strong effect of C on spike morphology, as was showed in Figure 1C, we could unambiguously discriminate in the RILs the compact from the normal spike by direct observation. The phenotypic data for SD was also not continuous (Figure 2) and showed a bimodal frequency distribution (lines 94-96 and Figure S1). Therefore, we could easily classify RILs as "Normal spike" and "Compact spike" groups. This classification is consistent with molecular markers analysis in the subsequent mapping results. We have added this results in lines 167-169.
Point 3: Figure 2 could be substituted by a graph on the frequency distribution of spike density. The bimodal distribution could suggest the genetic control by a major gene.
Response 3: Thank you for your suggestion. The data was showed in Figure S1 and this question is solved in “comments 1”
Point 4: “Linkage groups construction”. The paragraph should report the construction and description of the genetic map including in the map the examined gene Ppd-D1, Vrn-D1, Vrn-D3, Rht-B1b, and Rht8.
Response 4: Thank you for your suggestion. In this paragraph, we detected QTL for SD on 2D and for other morphological traits at the same locus, with the aim to report “the Pleiotropic effect of the Compactum gene”. In the next paragraph, we examined QTL for genes Ppd-D1, Vrn-D1, Vrn-D3, Rht-B1b, and Rht8 and studied the “combined effects of C gene with these genes”. Therefore, we think the present article structure would be better.
Point 5: “QTL detection for all the evaluated traits”. Report in the text and discuss all significant QTLs for all traits at LOD> 3.0. The QTL name in the text should always be complete with the trait abbreviation. Paragraphs 2.4 and 2.5 are not understandable for the reader as QTL names are reported without the character name (for example, what QTL does “nau-2D.2” refer to?).
Response 5: Thank you for the technical comments. We have revised it to “LOD> 3.0” in line 432, 437.
Sorry for the missing explanation of the abbreviation for these pleiotropic QTL. Take nau-2D.2 for example, while QPH.nau-2D.2/QSD.nau-2D.2/ QSL.nau-2D.2/ QTKW.nau-2D.2 are QTLs for four different traits, these QTLs are located within the same locus which had pleiotropic effects on PH, SL, SD and TKW. For convenience's sake, we designated this pleiotropic QTL as nau-2D.2 locus. We have added this explanation in lines 182-184, 285-286.
Point 6: Table and discussion on the classification of RILs according to phenotypic data on spike density and on molecular markers closely associated with the major QTL for spike density (SD) on 2D chromosome.
Response 6: Please see “comment 2 and reply”.
Point 7: Pleiotropic effects of SD gene/QTL with morpho-physiological and yield related trait.“Figure 4. Association of the QTLs with known genes located in the mapped QTL regions.” The figure shows the polymorphisms detected between the alleles of three genes and not the associations with QTLs.
Response 7: Sorry for the missing data. We have revised Figure 4 in the revised manuscript.
Point 8: Genes are not present in the genetic map.
Response 8: We have revised Figure 3 and added C gene in the genetic map.
Point 9: Relationship of mapped QTLs with known QTLs or genes. Report details on the statistical analysis.
Response 9: we have added details on how we predicted and validated the candidate genes in this paragraph.
Point 10: Report the abbreviations used in Figure S1, Figure 3 and Table 3.
Response 10: We have added a paragraph that defines the abbreviations at the end of the manuscript in lines 472-477 and added a footnote to the tables or figures.
Point 11: Discussion section: avoid concepts and data already reported in the results.
Response 11: We have deleted the contents appeared in the results in the revised manuscript.
Round 2
Reviewer 3 Report
The authors have attempted to respond to my earlier review but they are still not correct. My earlier review pointed out the the club wheat cultivar 'Bruehl' could not be the club parent because Bruehl has the Rht-B1b allele and therefore this locus could not be segregating if the pedigree of the RIL population was correct. In their response the authors stated that they had received six club wheat accessions from Washington State University in 1999 and that these were 'Madsen', 'Stephens", "Eltan", "Bruehl", "Hiller" and 'Rod". They further stated that the club wheat parent was actually not 'Bruehl' but rather 'Eltan' and that they had substituted Eltan for Bruehl throughout the manuscript. However, Madsen, Eltan, Stephens and Rod are not club wheat. They are all lax, soft white wheat. While both Bruehl and Hiller are club wheat, I already indicated why Bruehl can't be the parent. That leaves Hiller. Hiller is a semi-dwarf club wheat developed at WSU and Hiller has the Rht-D1b, rather than the Rht-B1b allele so it would segregate for both Rht alleles in your RIL population. However, there is no indication in the current manuscript that such is the case. In fact, the plant height listed in table 1 for the club wheat parent indicates to me that is does not possess a dwarfing allele and the scale on Supplementary figure 1 seems to indicate that plant height varied between 80 and 140 cm in the population. If both Rht loci were segregating, there should have been double dwarfs with plant heights below 60cm.
Another point that I missed in my earlier review. The photo of the parents in Figure 1 shows an Yangmai 153 with awns and the club wheat without awns. Both Bruehl and Eltan have awns so this trait would not segregate in the RIL population if either was the parent. Hiller does not have awns so it could be the parent, but the plant height is still not correct.
While I earlier stated that the paper was worthy of publication because it provides useful information, I am quite concerned that accurate records regarding the name of the two parents were apparently not kept. Also, it concerns me that the authors corrected their initial version by substituting the name of a lax wheat. Did they not grow these six lines in the field to see what they looked like?
I think you can see why I am concerned about the accuracy of other aspects of the paper. I really think that the authors need to obtain new sources of all six of the lines that they mention and conduct comparative phenotyping and genotyping to accurately identify the club wheat parent. And I think that the correct name should be used throughout the paper rather than the abbreviation TC.
Otherwise, the authors have addressed my earlier comments satisfactorily.
See these crop registration articles: Allan, R.E., C.J. Peterson, G.L. Rubenthaler, R.F. Line, and D.E. Roberts. 1989. Registration of ‘Madsen’ wheat. Crop Sci. 29(6):1575–1576. doi:10.2135/cropsci1989.0011183X002900060068x
Peterson, C.J., Jr., R.E. Allan, C.F. Morris, B.C. Miller, D.F. Moser, and R.F. Line. 1995. Registration of ‘Rod’ wheat. Crop Sci. 35(2):594.
Kronstad, W.E., C.R. Rhode, M.F. Kolding, and R.J. Metzger. 1978. Registration of ‘Stephens’ wheat. Crop Sci. 18(6):1097.
Peterson, C.J., Jr., R.E. Allan, G.L. Rubenthaler, and R.F. Line. 1991. Registration of ‘Eltan’ wheat. Crop Sci. 31(6):1704.
Peterson, C.J., Jr., C.F. Morris, R.F. Line, E. Donaldson, S.S. Jones, R.E. Allan. 1999. Registration of ‘Hiller’ Wheat. Crop Sci. 39:1531.
, , , , & (2001). Registration of ‘Bruehl’ wheat. Crop Science, 41, 2006–2007. https://doi.org/10.2135/cropsci2001.2006
Author Response
Response
Thanks, and we are sorry for making the confusion about the cultivar identity of the club wheat parent used for RIL population construction. We made the cross between Yangmai158 and club wheat, which was introduced in 1999 when Prof. Chen (our former colleague who is now retired from Nanjing Agricultural university) from Washington State University, USA. We communicated with Prof. Chen, he confirmed he introduced two club wheat cultivars Bruehl and Hiller. Therefore, as was pointed out by the reviewer, the club wheat parent which is awnless should be Hiller.
The reviewer also pointed out that Hiller should have Rht2 (Rht-D1b) gene, and the QTL would be detected and segregate in the RIL population. We searched references for the dwarfness of Hiller and only found one paper (Peterson, C.J., Jr., C.F. Morris, R.F. Line, E. Donaldson, S.S. Jones, R.E. Allan. 1999. Registration of ‘Hiller’ Wheat. Crop Sci. 39:1531), which was suggested by the reviewer. In this paper, the authors described that “It is a semidwarf putatively carrying the Rht2 gene.“ Except this, we did not find any other reports on whether Hiller has Rht2 or not.
To clarify this, we have cloned Rht2/rht2 from Yangmai158 and Hiller. We found their sequences are completely same, both carry the rht2 allele. In addition, there is no indication for the presence of Rht2 in our mapping result of the RIL population. Therefore, we propose that the Hiller does not have Rht2. This can explain why we did not detect Rht2 in our RIL population.
Reviewer 4 Report
Reviewer 4 (R4)
General comment on the revised manuscript.
Spike density or compact spike and its relationships with yield and yield components is a spike morphological trait widely studied since 1950 (Unrau 1950, Rao 1972, Gul and Allan 1972). This trait has been considered by wheat breeders to achieve greater number of kernels per spike and then to improve grain yield (Li et al., 2016). Chromosome location of the various spike density QTL/genes, including those on 2D chromosome, and the pleiotropic/associated effects on yield, yield components, spike morphology, plant height, etc. are well documented in several documents (Johnson et al., 2008; Manickavelu et al., 2011; Kowalski et al 2016; Zhai et al., 2016; Chai et al., 2018; Fan et al., 2019; Deng et al., 2019; Liu et al., 2020).
The merit of the manuscript could have been the precise localization of the major compact gene C on chromosome 2D and the investigation on the question whether the various traits associated with spike density can be attributed to pleiotropic effects of a single locus or to the joint effects of tightly linked genes/QTLs. This could be achieved with a high-density SNP based map integrated with some known genes, such as C, Rht8, Ppd1, and with an accurate QTL analysis for all the characters considered in the manuscript.
Response to Reviewer 4 Comments
Point 1:
R4 - The information must be presented in a simple and straightforward way to be understood by the reader.
Paragraph 2.1. on “Phenotypic analyses” should report the ANOVA of each trait for individual environments and across environments to ascertain significant differences between the RILs of the segregant population (Table could report in Supplemental Materials).
Response 1: We have performed ANOVA of each trait for individual environment and across environments, and obtained similar results to that from BLUP data. We have added this results in Supplemental file 1 and in lines 129-131.
R4 - Supplemental file 1 and lines 129-131 do not report the ANOVA of each trait for individual environments and across environments for the whole segregant population. It reports the comparison of the groups "Normal spike" vs "Compact spike". Moreover, the capture and the trait abbreviations are missing.
Point 2:
R4 - Table 1 “The 15 morphological traits in RIL population from the cross YM×TC”. As reported in the caption, the table should report data for the whole population without in this paragraph the separation in the two groups "Normal spike" and "Compact spike". Such classification based only on phenotypic data could be considered arbitrary and should be reported subsequently based also on data of the molecular markers closely associated with the major QTL for spike density (SD) on 2D chromosome.
Response 2: Thank you for the technical comments. We have reported data for the whole population in paragraph 4 under “2.1 Phenotypic analyses”. We have moved the paragraph 4 in front in lines 89-96.
R4 - Lines 89-96 do not report data for the whole population, such as mean, range, genetic variance, environmental variance, heritability, etc.
The statement “For all traits, significant genotypic effects were detected.” is not supported by data.
Response 2: Because of the strong effect of C on spike morphology, as was showed in Figure 1C, we could unambiguously discriminate in the RILs the compact from the normal spike by direct observation. The phenotypic data for SD was also not continuous (Figure 2) and showed a bimodal frequency distribution (lines 94-96 and Figure S1). Therefore, we could easily classify RILs as "Normal spike" and "Compact spike" groups. This classification is consistent with molecular markers analysis in the subsequent mapping results. We have added this results in lines 167-169.
R4 - The manuscript has the ambition to report the exact location of the C gene. The AAs report that “In this study, QTL analysis for SD was conducted to confirm whether the QTL position for compact spike phenotype could correspond to C locus.” (lines 201-202), and “By QTL mapping C gene was mapped to 2D within 370.12-406.29 Mb interval ……..” (lines 479-478).
Actually, by QTL mapping the AAs report the location of a QTL for SD (QSDnau-2D.1). AAs state that they could easily classify RILs as "Normal spike" and "Compact spike". Then they can code the two alternative phenotypes and integrate the gene C in the genetic map.
The assertion “This classification is consistent with molecular markers analysis in the subsequent mapping results.” is not supported by data.
My indication is that the classification of the lines should be carried out not only at the phenotypic level but also at the molecular level.
Point 3:
R4 - Figure 2 could be substituted by a graph on the frequency distribution of spike density. The bimodal distribution could suggest the genetic control by a major gene.
Response 3: Thank you for your suggestion. The data was showed in Figure S1 and this question is solved in “comments 1”.
R4 - I suggested the frequency distribution of spike density for each environment in the text in order to give higher relevance to the main topic of the manuscript (C gene and spike density).
Point 4:
R4 - “Linkage groups construction”. The paragraph should report the construction and description of the genetic map including in the map the examined genes Ppd-D1, Vrn-D1, Vrn-D3, Rht-B1b, and Rht8.
Response 4: Thank you for your suggestion. In this paragraph, we detected QTL for SD on 2D and for other morphological traits at the same locus, with the aim to report “the Pleiotropic effect of the Compactum gene”. In the next paragraph, we examined QTL for genes Ppd-D1, Vrn-D1, Vrn-D3, Rht-B1b, and Rht8 and studied the “combined effects of C gene with these genes”. Therefore, we think the present article structure would be better.
R4 - The analysis of the combined effects of C gene with Ppd-D1, Vrn-D1, Vrn-D3, Rht-B1b, and Rht8 genes” should be carried out with data of the examined RIL population and the related map. Maps obtained with different segregant populations can differ for the relative position of marker/genes. The AAs declare that they have analyzed these genes in the population and then they can integrate them in the map. In this way the combined effects can be analyzed with mapped loci in the same segregant population.
Lines 311-315 “The QPH.nau-2D.2/QSL.nau-2D.2/QSD.nau-2D.2/QTKW.nau-2D.2 were in the same locus which had pleiotropic effects on PH, SL, SD and TKW, respectively. The genetic interval of the locus (23.42-24.94 Mb) at the distal short arm of 2D overlaps with Rht8 gene (24.89 MB on 2DS) raising the possibility that Rht8 is the causative gene from YM.”
On what basis the authors affirm that ……” the locus (23.42-24.94 Mb) at the distal short arm of 2D overlaps with Rht8 gene…”? This objection also applies to all other genes.
Please check genetic and physical position of the detected QTL on 2D in Figure 3 and Table 2 and Table 3.
Point 5:
R4 - “QTL detection for all the evaluated traits”. Report in the text and discuss all significant QTLs for all traits at LOD> 3.0. The QTL name in the text should always be complete with the trait abbreviation. Paragraphs 2.4 and 2.5 are not understandable for the reader as QTL names are reported without the character name (for example, what QTL does “nau-2D.2” refer to?).
Response 5: Thank you for the technical comments. We have revised it to “LOD> 3.0” in line 432, 437.
Paragraphs 2.4 and 2.5 are not understandable for the reader as QTL names are reported without the character name (for example, what QTL does “nau-2D.2” refer to?).
Sorry for the missing explanation of the abbreviation for these pleiotropic QTL. Take nau-2D.2 for example, while QPH.nau-2D.2/QSD.nau-2D.2/ QSL.nau-2D.2/ QTKW.nau-2D.2 are QTLs for four different traits, these QTLs are located within the same locus which had pleiotropic effects on PH, SL, SD and TKW. For convenience's sake, we designated this pleiotropic QTL as nau-2D.2 locus. We have added this explanation in lines 182-184, 285-286.
R4 - The abbreviation for the pleiotropic QTL does not help the reader to read the paragraphs 2.4 and 2.5 easily. Furthermore, this abbreviation has no correspondence in Tables and Figure 3.
Point 6:
R4 -Table and discussion on the classification of RILs according to phenotypic data on spike density and on molecular markers closely associated with the major QTL for spike density (SD) on 2D chromosome.
Response 6: Please see “comment 2 and reply”.
R4 - Please see my objection to comment 2.
Point 7:
R4 - Pleiotropic effects of SD gene/QTL with morpho-physiological and yield related trait. “Figure 4. Association of the QTLs with known genes located in the mapped QTL regions.” The figure shows the polymorphisms detected between the alleles of three genes and not the associations with QTLs.
Response 7: Sorry for the missing data. We have revised Figure 4 in the revised manuscript.
R4 - The revised Figure 3 still has no date on pleiotropic effects of SD gene/QTL with morpho-physiological and yield related trait. The figure shows the polymorphisms detected between the alleles of three genes. The explanation of figures A, B and C is missing.
Point 8:
R4 - Genes are not present in the genetic map.
Response 8: We have revised Figure 3 and added C gene in the genetic map.
R4 - The revised Figure 3 reports the name of the SC QTLs and not the genes Ppd1 and Rht8 integrated in the map.
Point 9:
R4 - Relationship of mapped QTLs with known QTLs or genes. Report details on the statistical analysis.
Response 9: we have added details on how we predicted and validated the candidate genes in this paragraph.
R4 - The revised text does not report details on the statistical analysis.
Point 10:
R4 - Report the abbreviations used in Figure S1, Figure 3 and Table 3.
Response 10: We have added a paragraph that defines the abbreviations at the end of the manuscript in lines 472-477 and added a footnote to the tables or figures.
R4 - As also observed by the Reviewer 3, tables and figures should stand by themselves and not require additional referral to the text. The added footnote to the tables or figures still obliges the reader to refer to the text.
Point 11:
R4 - Discussion section: avoid concepts and data already reported in the results.
Response 11: We have deleted the contents appeared in the results in the revised manuscript.
Author Response
Please see the attachment. The last responses is hightlighted in red.
